# Elucidating the origins of multimode vibrational coherences of polyatomic molecules induced by intense laser fields

Zhengrong Wei[1], Jialin Li[1], Lin Wang[1], Soo Teck See[1], Mark Hyunpong Jhon[2], Yingfeng Zhang[3], Fan Shi[3], Minghui Yang[3] & Zhi-Heng Loh[1,4]

Strong-field laser–molecule interaction forms much of the basis for initiating and probing ultrafast quantum dynamics. Previous studies aimed at elucidating the origins of vibrational coherences induced by intense laser fields have been confined to diatomic molecules. Furthermore, in all cases examined to date, vibrational wave packet motion is found to be induced by $R$-selective depletion; wave packet motion launched by bond softening, though theoretically predicted, remains hitherto unobserved. Here we employ the exquisite sensitivity of femtosecond extreme ultraviolet absorption spectroscopy to sub-picometer structural changes to observe both bond softening-induced vibrational wave packets, launched by the interaction of intense laser pulses with iodomethane, as well as multimode vibrational motion of the parent ion produced by strong-field ionization. In addition, we show that signatures of coherent vibrational motion in the time-dependent extreme ultraviolet absorption spectra directly furnish vibronic coupling strengths involving core-level transitions, from which geometrical parameters of transient core-excited states are extracted.

[1] Division of Chemistry and Biological Chemistry, and Division of Physics and Applied Physics, School of Physical and Mathematical Sciences, Nanyang Technological University, Singapore 637371, Singapore. [2] Institute of High Performance Computing A*STAR, 1 Fusionopolis Way, #16-16 Connexis, Singapore 138632, Singapore. [3] Key Laboratory of Magnetic Resonance in Biological Systems, State Key Laboratory of Magnetic Resonance and Atomic and Molecular Physics, National Center for Magnetic Resonance in Wuhan, Wuhan Institute of Physics and Mathematics, Chinese Academy of Sciences, Wuhan 430071, China. [4] Centre for Optical Fibre Technology, The Photonics Institute, Nanyang Technological University, Singapore 639798, Singapore. Correspondence and requests for materials should be addressed to Z.-H.L. (email: zhiheng@ntu.edu.sg)

Femtosecond intense laser–matter interaction[1] has emerged as a powerful approach for unraveling the ultrafast quantum dynamics of molecules via processes such as high-order harmonic generation[2], above-threshold ionization[3], and laser-induced electron diffraction[4]. These phenomena have in turn been applied to, for example, the tomographic imaging of molecular orbitals[5, 6], the investigation of attosecond electronic wave packet dynamics[7–9], and the retrieval of transient molecular structures[10–12]. While many of these studies aim to elucidate electronic dynamics, it is important to realize that the intense laser–molecule interaction also inevitably triggers coherent vibrational motion, whose coupling to the electronic degrees of freedom will invariably affect the outcome and interpretation of experiments and theories that nominally seek to address electron dynamics[13–15]. Other factors that motivate the investigation of coherent vibrational dynamics in the strong-field regime include the prospect of employing intense laser fields to manipulate molecular dynamics[16–19] and the identification of ion electronic states produced by strong-field ionization by measuring their vibrational quantum beat frequencies[20].

Aside from the launching of vibrational wave packet dynamics in ions by strong-field ionization, the interaction of intense laser pulses with molecules can also generate vibrational coherences in the electronic ground state of neutral species. Two different mechanisms have been proposed: $R$-selective depletion[21] (Fig. 1a) and bond softening[22] (Fig. 1b). $R$-selective depletion arises from the dependence of the strong-field ionization rate on the nuclear coordinate, which yields, after depletion of the ground-state population by ionization, a vibrational probability density that is described by a non-stationary state. Within the quasi-static picture of bond softening, valid for low-frequency laser fields and high laser intensities, the field-induced distortion of the electronic ground-state potential drives the initially stationary vibrational probability density distribution away from its equilibrium position, hence initiating wave packet motion. Previous investigations of vibrational coherences induced on the electronic ground state by intense laser fields have identified $R$-selective depletion as the dominant mechanism for their generation[23–25]. In pioneering studies by Ullrich et al., for example, an initial cosinusoidal oscillation phase of $(0.946 \pm 0.001)\pi$ rad in the temporal evolution of the $D_2$ bond length suggests the launching of the vibrational wave packet from the outer turning point of the $D_2$ potential[23]. This observation is consistent with the larger ionization potential, and hence, lower ionization rate, for D–D bond lengths larger than the equilibrium bond length. Cosinusoidal

phases approaching 0 mod $\pi$, indicative of $R$-selective depletion, are also observed for vibrational coherences generated in the electronic ground states of $I_2$[24] and $Br_2$[25]. These studies beget the question as to whether vibrational wave packets can be created by bond softening. Furthermore, the generality of launching vibrational wave packets in the electronic ground state of polyatomic molecules by intense laser pulses remains to be established. The ability to harness intense laser fields to drive vibrational coherences would complement nonresonant impulsive stimulated Raman scattering, which operates in the perturbative limit at relatively low laser intensities[26].

Here we employ femtosecond extreme ultraviolet (XUV) absorption spectroscopy[27, 28] to elucidate the origins of multi-mode coherent vibrational motion induced by intense laser fields in neutral iodomethane ($CH_3I$) and its parent ion ($CH_3I^+$). XUV absorption spectroscopy is uniquely suited to the present investigation due to its ability to distinguish, with electronic state resolution, the iodine (I) $4d$ core-level transitions of $CH_3I$ and $CH_3I^+$, as well as those of the atomic photofragments I and $I^+$. In the case of the $CH_3I$ $\tilde{X}\ ^1A_1$ electronic ground state, analysis of the spectral modulations induced by C–I stretching motion ($\nu_3$) yields the relative contributions of bond softening and $R$-selective depletion to the generation of the wave packet (Fig. 1c). In addition, wave packet motions along the C–I stretch ($\nu_3$) and $CH_3$ umbrella ($\nu_2$) modes are observed in the $CH_3I^+$ $\tilde{X}^+\ ^2E_{3/2}$ parent ion. The results unambiguously show the existence of bond softening-induced wave packets and demonstrate the exquisite sensitivity of femtosecond XUV absorption spectroscopy to structural changes on the sub-picometer length scale. Furthermore, we show that XUV absorption probing of coherent vibrational dynamics allows the retrieval of structural parameters for transient core-excited states.

## Results

**Experimental results**. The differential XUV absorption spectra collected as a function of pump-probe time delay is shown in Fig. 2a (see "Methods" and "Supplementary Methods" sections for details of the experimental set-up). The XUV absorption features in this energy range are assigned to transitions from the spin-orbit-split I $4d$ core level to the unoccupied valence orbitals with predominantly I $5p$ character. Pronounced spectral modulations as a function of time delay are observed in the region of the neutral $CH_3I$ ground-state bleach transitions (~50–53 eV) (see Supplementary Note 1), i.e., the I $4d_j \rightarrow \sigma^*_{C–I}$ transitions[29]

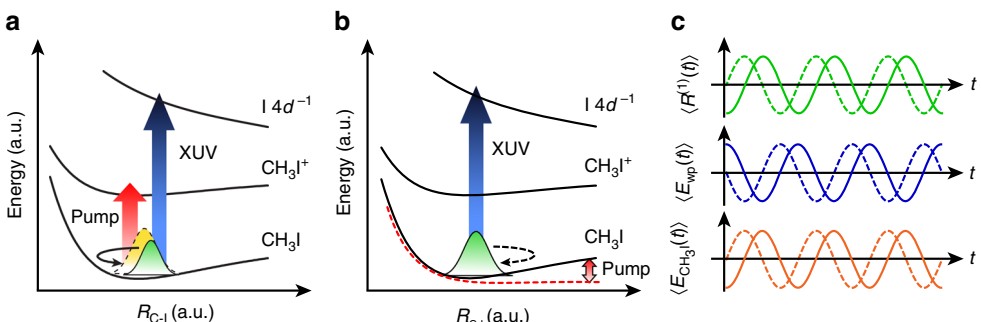

**Fig. 1** Generation of vibrational wave packets by intense laser fields. Schematic illustration of mechanisms of vibrational wave packet generation along the C–I stretching mode in the electronic ground state of $CH_3I$. An intense near-infrared (NIR) pump pulse initiates wave packet motion by **a** $R$-selective depletion and **b** bond softening, which is subsequently probed by extreme ultraviolet (XUV) absorption. The *yellow curve* in **a** depicts the equilibrium vibrational probability density, whereas the *green curves* in both **a**, **b** represent the vibrational probability density of the wave packet launched by the intense laser field. **c** shows the subsequent time-dependent position of the vibrational wave packet $\langle R^{(1)}(t) \rangle$, its XUV transition energy $\langle E_{wp}(t) \rangle$, and the spectral first-moment signal $\langle E_{CH_3I}(t) \rangle$. The *solid* and *dashed lines* correspond to the trajectories for vibrational wave packets induced by $R$-selective depletion and bond softening, respectively

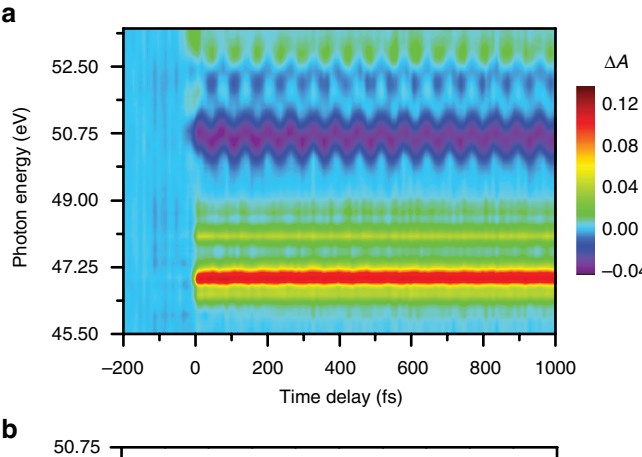

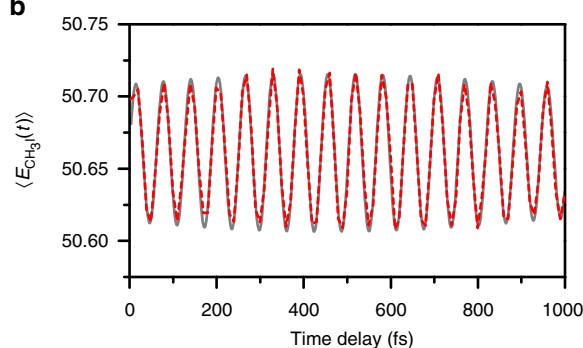

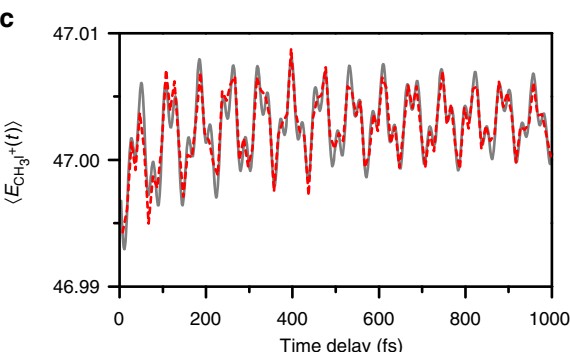

**Fig. 2** I $4d$ transient absorption spectra of stong-field-ionized CH$_3$I. **a** Contour plot of the differential XUV absorption spectra collected as a function of pump-probe time delay. Time-resolved spectral first-moments calculated about the **b** neutral CH$_3$I $4d_j \rightarrow \sigma^*_{C-I}$ resonance, $\langle E_{CH_3I}(t) \rangle$, and the **c** CH$_3$I$^+$ I $4d_{5/2} \rightarrow e_{3/2}^{-1}$ resonance, $\langle E_{CH_3I^+}(t) \rangle$, are also shown. The *dashed red lines* are the data and the *solid gray lines* are obtained from fits

($j = 3/2$ or $5/2$). The positive differential absorption signals observed in the ~46–49 eV range originate from the CH$_3$I$^+$ parent ion and the atomic I and I$^+$ photoproducts of dissociative ionization[30]. The prominent peak centered at 47.0 eV can be assigned to the I $4d_{5/2} \rightarrow e_{3/2}^{-1}$ transition of the CH$_3$I$^+$ $\tilde{X}^+$ $^2E_{3/2}$ parent ion, based on the energetic difference between the binding energies of the I $4d_{5/2}$ core level[31] and the $e_{3/2}$ valence level[32] ($E_{4d_{5/2}} - E_{e_{3/2}} = 47.08$ eV). Analysis of the appearance times of I and I$^+$ reveals ultrafast dissociative ionization dynamics, which will be detailed in a subsequent report. Here we focus on the vibrational wave packet dynamics encoded in the I $4d_j \rightarrow \sigma^*_{C-I}$ and $4d_{5/2} \rightarrow e_{3/2}^{-1}$ transitions of CH$_3$I and CH$_3$I$^+$, respectively.

**Vibrational wave packet generation mechanism in neutral CH$_3$I.** The time-dependent spectral first-moment $\langle E_{CH_3I}(t) \rangle$ computed about the I $4d_j \rightarrow \sigma^*_{C-I}$ transition of neutral CH$_3$I reflects strong-field-induced wave packet motion along the C–I

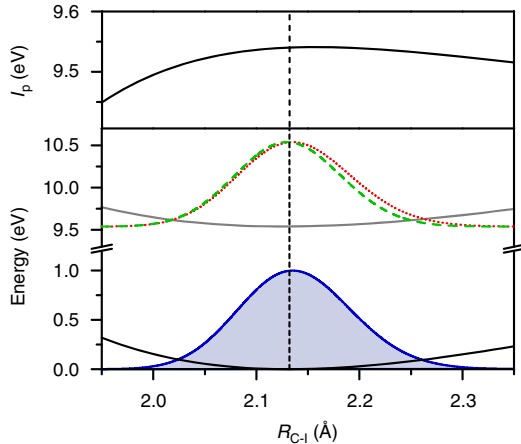

**Fig. 3** Launching of sub-picometer scale C–I stretching motion in CH$_3$I$^+$. The vertical ionization potential $I_p$ of CH$_3$I as a function of C–I bond distance (*top panel*), as obtained from the energy gap between the Morse potentials of CH$_3$I (*solid black line*) and CH$_3$I$^+$ (*solid gray line*) along the C–I stretch coordinate (*bottom panel*). The spectroscopic parameters used to construct the potentials are given in Supplementary Note 6. The *vertical dashed line* denotes the equilibrium C–I bond distance of CH$_3$I. The vibrational probability densities for CH$_3$I, initially at thermal equilibrium (*solid blue line*), the CH$_3$I$^+$ fraction produced by ionization (*red dotted line*), and the CH$_3$I$^+$ $v = 0$ eigenstate (*green dashed line*) are also shown

stretching mode in its electronic ground state (Fig. 2b). Beyond the high-frequency oscillation, the presence of an envelope modulation suggests a fit to two oscillatory components, i.e., $\langle E_{CH_3I}(t) \rangle = \sum \langle E_{CH_3I}^{(i)}(0) \rangle \cos[\omega_{CH_3I}^{(i)} t + \phi_{CH_3I}^{(i)}]$ ($i = 1, 2$). The first component, with an initial phase of $\phi_{CH_3I}^{(1)} = -(0.49 \pm 0.05)\pi$ rad, is consistent with bond softening-induced wave packet motion. Because the I $4d_j \rightarrow \sigma^*_{C-I}$ transition energy monotonically decreases with C–I bond length $R_{C-I}$ (see Supplementary Note 2), it can be inferred that an initial displacement toward larger $R_{C-I}$ leads to $\phi = -\pi/2$ rad (see Supplementary Notes 3 and 4); conversely, initial displacement toward smaller $R_{C-I}$ leads to $\phi = \pi/2$ rad. Within the Floquet picture, the one- and two-photon dressing of the neutral ground state by the dissociative $^3Q_{0^+}(A_1)$ state[19, 33] results in the softening of the ground-state C–I potential (see Supplementary Note 5). In the quasi-static limit of laser–molecule interaction, the intense laser field distorts the neutral ground state potential along the C–I coordinate. Both pictures yield the same outcome: the equilibrium vibrational probability density is initially displaced toward larger $R_{C-I}$ by the intense laser field, hence launching wave packet motion that is characterized by $\phi_{CH_3I}^{(1)} = -\pi/2$ rad (Fig. 1a). The excellent agreement between the oscillation frequency $\omega_{CH_3I}^{(1)} = 533 \pm 1$ cm$^{-1}$ and the literature value for the fundamental $\nu_3$ frequency[34] (533 cm$^{-1}$) is consistent with the bond softening-induced wave packet motion being dominated by the coherent superposition of the vibrational ground state and the first-excited state.

The second component of $\langle E_{CH_3I}(t) \rangle$ has an initial phase of $\phi_{CH_3I}^{(2)} = -(0.09 \pm 0.12)\pi$ rad, indicative of wave packet motion launched by $R$-selective depletion from the outer turning point of the C–I potential (Fig. 1a) (see Supplementary Note 4). This observation can be rationalized by inspecting, in the vicinity of the equilibrium $R_{C-I}$ for CH$_3$I, the energy gap between the spectroscopically reconstructed Morse potentials of the CH$_3$I and CH$_3$I$^+$ electronic ground states as a function of $R_{C-I}$ (Fig. 3, *top panel*; see Supplementary Note 6 for the parameters of the Morse potentials). This energy gap corresponds to the vertical ionization potential of CH$_3$I. Its steep decrease for C–I bond lengths shorter

than the equilibrium value favors strong-field ionization at the inner turning point of the C–I potential, hence launching wave packet motion from the outer turning point with a $\phi^{(2)}_{CH_3I}$ value of 0 rad. The measured oscillation frequency of $\omega^{(2)}_{CH_3I} = 518 \pm 3$ cm$^{-1}$, anharmonically shifted from the fundamental frequency of 533 cm$^{-1}$, suggests a vibrational coherence that involves mainly the $v = 2$ and $v = 3$ levels of the $v_3$ mode[35], whose estimated fractional populations at a vibrational temperature of 353 K are 0.0117 and 0.0014, respectively. The high-lying vibrational states dominate the $R$-selective depletion response because they possess higher probability densities near the turning points of the C–I potential, where strong-field ionization rates are larger.

**Multimode coherent vibrational motion in CH$_3$I$^+$.** Aside from the C–I stretching wave packet initiated in neutral CH$_3$I by the intense laser field, the time-dependent spectral first-moment $\langle E_{CH_3I^+}(t) \rangle$ computed about the dominant CH$_3$I$^+$ I $4d_{5/2} \to e^{-1}_{3/2}$ resonance at 47.0 eV reveals oscillatory components that can be assigned to the C–I stretching ($v_3$) and CH$_3$ umbrella ($v_2$) modes of the $\tilde{X}^+$ $^2E_{3/2}$ state (Fig. 2c). The $\langle E_{CH_3I^+}(t) \rangle$ time trace can be fit to a sum of two damped cosinusoidal oscillations, i.e., $\langle E_{CH_3I^+}(t) \rangle = \sum \langle E^{(i)}_{CH_3I^+}(0) \rangle \cos[\omega^{(i)}_{CH_3I^+} t + \phi^{(i)}_{CH_3I^+}] e^{-t/\tau^{(i)}_{CH_3I^+}}$ ($i = 1, 2$). The retrieved vibrational frequencies, $\omega^{(1)}_{CH_3I^+} = 1253 \pm 1$ and $\omega^{(2)}_{CH_3I^+} = 475 \pm 1$ cm$^{-1}$, match those reported for the $v_2$ (1257 cm$^{-1}$) and $v_3$ (480 cm$^{-1}$) modes, respectively[36]. The vibrational wave packets for both modes dephase with comparable damping times: $\tau^{(1)}_{CH_3I^+} = 1.8 \pm 0.6$ ps and $\tau^{(2)}_{CH_3I^+} = 1.7 \pm 0.4$ ps. The simultaneous appearance of both $v_2$ and $v_3$ modes in the vibrational wave packet is noteworthy, given that previous studies of polyatomic molecular ions produced by strong-field ionization resolved wave packet motion only along a single vibrational coordinate[37, 38].

The initial oscillation phase of $\phi^{(1)}_{CH_3I^+} = (0.13 \pm 0.12)\pi$ rad for the $v_2$ mode suggests displacive excitation of the CH$_3$ umbrella mode wave packet. Displacive excitation occurs when the Franck–Condon region accessed by vertical ionization is displaced from the equilibrium geometry, such that the ensuing structural rearrangement that accompanies equilibration leads to wave packet motion with an initial phase of 0 mod $\pi$ rad[39]. In this case, displacive excitation of the $v_2$ mode is supported by the small, but nevertheless, significant difference in the H–C–I bond angles between the neutral[39] and ion[40, 41] species: 107.7° in CH$_3$I vs. 108.3° in CH$_3$I$^+$. Interestingly, the measured oscillation phase $\phi^{(2)}_{CH_3I^+}$ of $(0.71 \pm 0.05)\pi$ rad for the C–I stretching mode deviates, beyond error, from phase values that would be expected for wave packet dynamics induced by either displacive excitation ($\pi$ rad) or bond softening ($\pi/2$ rad). One plausible explanation is the simultaneous launching of wave packets by both displacive excitation and bond softening, such that the overall oscillatory signal at $\omega^{(2)}_{CH_3I^+} = 475$ cm$^{-1}$ can be expressed as: $\cos[\omega^{(2)}_{CH_3I^+} t + \phi^{(2)}_{CH_3I^+}] = \sin\phi^{(2)}_{CH_3I^+} \cos[\omega^{(2)}_{CH_3I^+} t + \pi/2] - \cos\phi^{(2)}_{CH_3I^+} \cos[\omega^{(2)}_{CH_3I^+} t + \pi]$, where the terms with amplitude coefficients $\sin\phi^{(2)}_{CH_3I^+}$ and $-\cos\phi^{(2)}_{CH_3I^+}$ arise from bond softening and displacive excitation, respectively. A phase value of $\phi^{(2)}_{CH_3I^+} = (0.71 \pm 0.05)\pi$ rad gives an amplitude ratio of $-\sin\phi^{(2)}_{CH_3I^+} / \cos\phi^{(2)}_{CH_3I^+} = 1.3 \pm 0.1$, which in turn suggests that bond softening dominates over displacive excitation. It is conceivable that the same intense laser pulse that drives strong-field ionization can also induce the softening of the C–I potential of the resultant CH$_3$I$^+$ ions. Displacive excitation, on the other hand, is supported by the different equilibrium C–I bond lengths

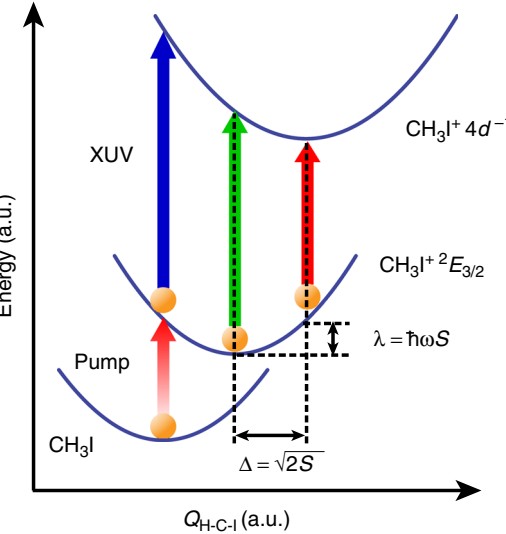

**Fig. 4** Retrieval of the equilibrium H-C-I angle for the I $4d$ core-excited state. Schematic illustration of potentials along the CH$_3$ umbrella ($v_2$) mode for CH$_3$I, CH$_3$I$^+$ ($\tilde{X}^+$ $^2E_{3/2}$), and CH$_3$I$^+$ (I $4d^{-1}_{5/2}$). The amplitude of the component of $\langle E_{CH_3I^+}(t) \rangle$ that oscillates at the frequency $v_2$ can be used to retrieve the Huang–Rhys factor $S$ and the dimensionless displacement $\Delta$ between the two CH$_3$I$^+$ states, and hence, determine the equilibrium H-C-I angle for the I $4d^{-1}$ core-excited CH$_3$I$^+$ species

of CH$_3$I (2.132 Å)[40] and CH$_3$I$^+$ (2.126 Å)[41]. $R$-selective depletion calculations that consider averaging over both thermal and laser intensity distributions confirm that the initial vibrational probability density created in the CH$_3$I$^+$ $\tilde{X}^+$ $^2E_{3/2}$ state indeed extends to larger $R_{C-I}$ compared to that of the $v = 0$ eigenstate (Fig. 3, bottom panel; see Supplementary Note 7 for details of simulations). It is remarkable that the decrease in equilibrium C–I bond length of 0.006 Å, representing a mere change of 0.3%, can yield a discernible modulation of the XUV transition energy. For comparison, vibrational wave packets[42] or coherent phonons[39] detected by optical pump-probe spectroscopy typically involve changes in bond lengths of a few percent. The enhanced sensitivity of femtosecond XUV absorption spectroscopy to structural changes originates from the highly repulsive nature of the core-hole-excited state, in this case, the I $4d^{-1}_{5/2}$ state of CH$_3$I$^+$ (see Supplementary Note 2). Access to such a repulsive potential upon absorption of the XUV probe pulse allows small geometrical distortions to be mapped onto measurable shifts in XUV transition energies.

**Retrieval of I $4d^{-1}$ core-excited state geometries.** Aside from unraveling the mechanism of wave packet generation, quantitative analysis of the time-resolved XUV absorption spectra also provides insight into the nature of the I $4d^{-1}_{5/2}$ core-excited state. The strength of the vibronic coupling between the CH$_3$I$^+$ I $4d_{5/2} \to e^{-1}_{3/2}$ XUV probe transition and the $v_2$ vibrational mode, as characterized by the Huang–Rhys factor[43] $S$, is recovered from the oscillation amplitude via the relation $\langle E^{(1)}_{CH_3I^+}(0) \rangle = 2\lambda = 2\hbar\omega^{(1)}_{CH_3I^+} S$, where $\lambda$ is the reorganization energy[44] (Fig. 4). (Note that the dissociative nature of the I $4d^{-1}_{5/2}$ core-excited state potential along the C–I stretch coordinate precludes the same analysis from being performed for the $v_3$ mode.) The amplitude $\langle E^{(1)}_{CH_3I^+}(0) \rangle = (2.6 \pm 0.2) \times 10^{-3}$ eV therefore yields a Huang–Rhys factor of $(8.3 \pm 0.7) \times 10^{-3}$, signifying weak vibronic coupling. Furthermore, combining the dimensionless displacement of $\Delta = \sqrt{2S} = 0.13 \pm 0.01$ with the measured phase $\phi^{(1)}_{CH_3I^+}$ of ~0 rad and the experimentally determined H–C–I angle of CH$_3$I$^+$ in the $\tilde{X}^+$ $^2E_{3/2}$ state[41]

$(108.3 \pm 0.3)°$ yields an equilibrium H–C–I angle of $(110.0 \pm 0.6)°$ for the I $4d_{5/2}^{-1}$ core-excited state of $CH_3I^+$ (see Supplementary Note 8).

## Discussion

The emergence bond softening-induced coherent vibrational motion in the present work contrasts with earlier reports, all of which observed vibrational wave packet dynamics that are attributed to R-selective depletion[23–25]. In those cases, strong-field ionization results from the removal of either a bonding ($D_2$) or an anti-bonding electron ($Br_2$ and $I_2$). The change in the formal bond order upon ionization is accompanied by pronounced changes in the equilibrium bond length and vibrational frequency. For example, the equilibrium bond length[45] of $D_2$ upon ionization increases from 0.7415 to 1.0559 Å and its vibrational frequency decreases[46, 47] from 2993.60 to 1577.17 $cm^{-1}$. These dramatic changes in the relative displacement and curvatures of the neutral and ion potential energy curves yield an ionization potential that is strongly dependent on bond length, hence favoring the generation of vibrational wave packets via R-selective depletion. In the case of $CH_3I$, however, strong-field ionization to the ion $\tilde{X}^+$ state corresponds to the removal of an electron from the nonbonding iodine $5p$ valence orbital[48]. The C–I potential curves for neutral and ion species therefore run largely parallel to one another (see Supplementary Note 6), as evidenced by the subtle differences in the experimental C–I bond lengths (2.132 Å for $CH_3I$ vs. 2.126 Å for $CH_3I^+$)[40, 41] and $\nu_3$ frequencies (533 $cm^{-1}$ for $CH_3I$ vs. 480 $cm^{-1}$ for $CH_3I^+$)[34, 36], as well as the vanishingly small $\nu_3$ vibrational progression in the $\tilde{X}^+ \, ^2E_{3/2}$ band of the $CH_3I$ photoelectron spectrum[49]. The strongly suppressed R-selective depletion channel brings to fore the contribution from bond softening. Along these lines, one would also expect the efficient launching of vibrational wave packets by bond softening in other molecules for which valence ionization results in the removal of an electron from a nonbonding orbital. Examples of such molecules include some that are commonly used in strong-field investigations, e.g., $HCl$[4], $CO_2$[38], $H_2O$[50], alcohols[51], and organic halides[19]. Since bond softening corresponds to far-off-resonance Raman scattering within the photon picture[23], vibrational modes that are Raman active are expected to be susceptible to bond softening.

The amplitudes of vibrational motion launched by bond softening and R-selective depletion should each exhibit a distinct dependence on the laser intensity. In general, bond softening is expected to dominate at lower intensities because of limited ionization depletion of the neutral ground state. In the case of $D_2$, model calculations[23] reveal that the relative contributions are reversed at a threshold intensity of $I_{th} \sim 2.5 \times 10^{14}$ W $cm^{-2}$, with R-selective depletion dominating the response for intensities $I > I_{th}$. Whether such an intensity threshold exists for $CH_3I$ is unclear, given the negligible displacement between the equilibrium C–I distances between the ground-state neutral and ion species ($\Delta R / R = 0.003$) and the similarity between the vibrational frequencies, and hence, the curvatures of the potentials of the two states ($\Delta \omega / \omega = 0.1$). In the extreme limit where both $\Delta R$ and $\Delta \omega / \omega$ are zero, R-selective depletion vanishes and only bond softening-induced wave packets can exist. In such cases, it is interesting to note that strong-field ionization to higher-lying electronically excited ion states could still result in vibrational motion launched by R-selective depletion. Future experiments that determine the relative contributions from bond softening and R-selective depletion as a function of laser intensity should allow one to address if an intensity threshold for $CH_3I$ exists and if electronically excited ion states participate in R-selective depletion.

The determination of equilibrium geometries of core-excited states is a non-trivial task. Experimentally, vibronic progressions that appear in static core-level photoabsorption spectra are analyzed to yield Franck–Condon factors, from which the equilibrium geometries of the N $1s^{-1}$ core-excited $N_2$ and Si $2p^{-1}$ core-excited $SiH_4$ have been inferred, e.g.,[52, 53]. However, this method is limited to core-hole states with sufficiently long lifetimes, whose narrow linewidths support clearly resolved vibronic features. For example, the natural linewidths of N $1s^{-1}$ and Si $2p^{-1}$ are 135 and 50 meV, respectively[52, 53], which are comparable to or smaller than the vibrational energy spacings in the vibronic progression. This frequency-domain approach clearly cannot be extended to low-frequency vibrational modes and/or short-lived core-excited states, such as the I $4d^{-1}$ core levels probed herein (FWHM $\sim$0.5 eV)[54]. Ab initio calculations of core-excited equilibrium geometries are similarly challenging due to the highly excited nature of the core-hole states[55, 56], making them beyond the reach of electronic structure methods, such as multi-reference configuration interaction and complete active-space self-consistent field, that are typically employed to determine the equilibrium geometries of valence-excited states. In the present work, we show that time-domain XUV probing of vibrational wave packet dynamics furnishes the vibronic coupling involving the core-level transition and hence, the dimensionless displacement parameter, which can in turn be used to recover the equilibrium geometry of the core-excited state. While our analysis is performed only for the $\nu_2$ umbrella mode, it is important to note that the approach presented herein can be broadly generalized to the core-excited states of other molecules along their various vibrational coordinates. The ability to recover the molecular geometries of core-excited states will benefit the extensive efforts that are being pursued at X-ray free-electron lasers to investigate the ultrafast dynamics triggered by core-level excitation[57, 58].

Finally, we note that previous studies of vibrational wave packet dynamics in polyatomic molecular ions created by strong-field ionization have elucidated vibrational motion only along one dimension[37, 38]. The observation of strong-field-induced multi-mode vibrational coherences, such as those reported here for $CH_3I^+$ in the $\tilde{X}^+ \, ^2E_{3/2}$ state, is unprecedented and can be attributed to multiple advantages offered by our experimental approach. First, XUV absorption probing permits the spectroscopic identification of both the chemical species that are produced by strong-field ionization as well as their electronic states[27]. Second, large shifts in the core-level transition energies accompany small structural changes, thus enabling spectrally resolved XUV absorption spectroscopy to detect coherent vibrational amplitudes on the sub-picometer scale, as demonstrated in the present work. Third, XUV absorption spectroscopy offers exquisite time resolution, extendable to the attosecond time scale[59], which translates to a large detection bandwidth for coherent vibrational motion. Fourth, the use of few-cycle pulses for strong-field ionization effectively drives wave packet dynamics involving a large manifold of vibrational modes and also provides a temporally well-defined trigger for the wave packet dynamics. The latter is crucial for establishing the initial phases of the wave packet oscillations and hence, the mechanisms for generating the vibrational coherences. Altogether, these factors point to femtosecond XUV absorption as a powerful technique for unraveling the multimode vibrational dynamics of transient molecular ions produced by intense laser–molecule interaction.

## Methods

The experimental set-up (see "Supplementary Methods" section for a detailed description) employs 0.8-mJ, 5.6-fs pulses centered at 786 nm to drive high-order harmonic generation in argon gas. The resultant XUV light is transmitted through a 0.2-μm-thick aluminum foil, hence removing the residual near-infrared (NIR),

before it is refocused by a gold-coated toroidal mirror onto the sample gas target. The sample target is a 3-mm-path length quasi-static gas cell that is heated to 353 K. The $CH_3I$ vapor pressure in the gas cell is 14 mbar. The transmitted XUV radiation is spectrally dispersed onto an X-ray CCD camera by a flat-field XUV grating. The strong-field-ionizing NIR pump pulse has a peak intensity of $1.9 \times 10^{14}$ W cm$^{-2}$ and intersects the XUV probe beam at the sample target with a crossing angle of 1°. A variable time delay between the NIR pump and XUV probe pulses is introduced by means of a computer-controlled piezo-driven delay stage positioned in the path of the NIR pump beam. The midpoint of the rise in the $CH_3I^+$ ($\tilde{X}^+ \, ^2E_{3/2}$) parent ion signal at 47.0 eV, when the instantaneous ionization rate is the highest, is used to define time-zero to a precision of 1.2 fs. The typical signal-to-noise ratio achieved after averaging over 15 scans is ~20:1.

**Data availability**. The data that supports the findings of this study are available from the corresponding author on request.

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

## Acknowledgements

This work is supported by a NTU start-up grant, the A*Star Science and Engineering Research Council Public Sector Funding (122-PSF-0011 and 122 360 0008), the Ministry of Education Academic Research Fund (MOE2014-T2-2-052), and the award of a Nanyang Assistant Professorship to Z.-H.L. M.Y. acknowledges the financial support from National Natural Science Foundation of China (NSFC, Project No. 21373266). We are grateful to K. Yamanouchi, T. Kobayashi, S.L. Chin, and D. Mathur for useful discussions.

## Author contributions

Z.W. and J.L. performed the experiment. M.H.J., Z.-H.L., Y.Z., F.S., and M.Y. performed the theoretical calculations. J.L., Z.W., L.W., S.T.S., and Z.-H.L. designed and constructed the experimental set-up. Z.W. and Z.-H.L. wrote the manuscript, with input from all the authors.

## Additional information

**Competing interests:** The authors declare no competing financial interests.

