## [Peer Review File · Nature Communications]

Reviewers' comments:

Reviewer #1 (Remarks to the Author):

At the onset, let me say that I have been working in the ultrafast laser spectroscopy field for nearly twenty years. This is the most excitingly new, and rigorously thorough paper I have read on the topic in a very long time. The work has the potential for making a great impacts within and beyond the ultrafast spectroscopy community. I strongly support publication of this work in Nature Communications with only a few suggested revisions. In fact, I would advocate publishing this manuscript in Nature.

The authors describe the mechanisms by which vibrational wavepackets are formed in polyatomic molecules. The significant outcomes of this work include: 1) the first unambiguous experimental verification of vibrational wavepackets formed by bond softening mechanisms; 2) as far as I can tell, the first observation of vibrational wavepackets formed on the ground state of a polyatomic molecule – this is important because it provides access to nuclear motion on multiple potential surface, as opposed to the 1-D examinations of prior studies on diatomic molecules; and 3) the quantification of the relative signal amplitudes for competitive excitation channels (i.e. R-type vs. bond softening). It is important to appreciate that the bond-softening mechanism has been proposed, but not experimentally verified, for some time. It is likely that this mechanism has been a contributor to signals measured and reported by many experimentalists, but conventional spectroscopies do not have the needed resolution to unambiguously identify this source; it requires the ultrafast EUV probing capabilities unique to the authors. These experimental data provide the “smoking gun” evidence needed to assign the bond softening mechanism – the $-\pi/2$ phase shift observed in the oscillatory time-dependent transient signals. In addition to its importance to fundamental molecular dynamics research, these finding also have the potential to impact areas such as analytical chemistry by enabling a new type of polyatomic vibrational spectroscopy.

Specific suggestions for revision.

1. As I detail above, this is an important paper, and there is little room for argument over the data interpretation – the authors got it right. However, I don't think the title accurately reflects the content of the manuscript, nor will it attract the broadest readership. In my opinion, the novelty of this work is the detailed assessment of polyatomic vibrational wavepackets and their excitation mechanism. Therefore, I would encourage modifying the title to explicitly state polyatomic molecules are being studied. I know this is more or less implicit in the use of multimode, but I don't know that the average chemist will be attracted by this title. I encourage more specificity here.

2. Figures S7 and S8 are critically important to understanding the mechanisms at play. I encourage modifying Figure 1 of the main manuscript to include the more descriptive elements of Figs. S7 and S8. Again, this will help a broader cross section of readers to understand the manuscript.

3. This reviewer is very curious about the possibility to access exotic regions of the potential surface, or modes with different symmetries by using light fields with different polarization states. Have the authors considered how the results might be influenced using circularly polarized light vs. linearly polarized light? Can this be commented on in the manuscript?

4. I do make one recommendation that I think can strengthen the conclusions reached by the authors. An important aspect of this work is their ability to resolve the relative contributions of two

mechanisms for exciting vibrational wavepackets (R-selective depletion vs. bond softening). I really want to emphasize that this capability really sets the current manuscript apart from any others I've seen on the topic. But, this capability presents a great opportunity. Because the mechanisms should seem to have an excitation field dependence, the authors could consider analyzing how the relative amplitudes respond over a range of pulse energies. It would be very helpful to identify if there are thresholds for these different mechanisms and what molecular parameters set these parameters. The authors could expand the discussion on pages 6-8 to include this analysis.

5. Related to comment 4, on page 3 the authors state that bond-softening is valid for low-frequency laser fields. What is "low-frequency" in this context? Again, I am very interested in the relationship between frequency and excitation field strength. Are these "low frequencies" likely to be molecule or symmetry specific? Understanding these limits are important for the development of applications in vibrational spectroscopy.

6. Finally, can the authors comment on how general the bond softening mechanism might be. Depletion methods are easy to understand, but is the bond softening mechanism limited to specific ground state potentials? Can we expect this method to be applicable to molecules beyond halogen-containing species?

Reviewer #2 (Remarks to the Author):

This is a beautiful manuscript, which presents results on using time resolved XUV absorption to track vibrational wave packets launched by a strong field laser pulse. The results are very nice and the analysis and discussion thorough and enlightening.

At first I was somewhat skeptical of the results, because it seemed to me that the sensitivity of the measurement to displacements claimed by the authors was not possible. However, I went through the analysis in detail, and given the signal to noise and sensitivity of the authors to small changes in the spectrally resolved absorption with time delay as well as the large shift in the absorption spectra with displacement, it indeed seems like it is possible to achieve the sensitivity to small displacements that the authors claim.

I was also a little skeptical of the extraction of the phase information for each of the modulation frequencies, but the SI gives a nice detailed discussion of this which I found convincing. I found the discussion of why the authors find a different amount of bond softening vs R dependent ionization from earlier results very nice and informative. However, I think it would be good to have some discussion of the intensity dependence of their measurements. How do the pump probe measurements vary with pump pulse intensity?

While the authors here focus on the measurement of bound state vibrational wave packets, which have been studied with many other approaches, I feel that the sensitivity they achieve and the potential for using their approach to study a broader class of wave packets merits publication in Nature Communications.

So, in conclusion, I think that this is a very well written and exciting manuscript which demonstrates an exquisite sensitivity to vibrational motion via transient XUV absorption spectroscopy. I believe it deserves to be published in Nature Communications once the authors add some discussion of the pump pulse intensity dependence.

Response to reviewers

Reviewer #1 (Remarks to the Author):

At the onset, let me say that I have been working in the ultrafast laser spectroscopy field for nearly twenty years. This is the most excitingly new, and rigorously thorough paper I have read on the topic in a very long time. The work has the potential for making a great impacts within and beyond the ultrafast spectroscopy community. I strongly support publication of this work in Nature Communications with only a few suggested revisions. In fact, I would advocate publishing this manuscript in Nature.

The authors describe the mechanisms by which vibrational wavepackets are formed in polyatomic molecules. The significant outcomes of this work include: 1) the first unambiguous experimental verification of vibrational wavepackets formed by bond softening mechanisms; 2) as far as I can tell, the first observation of vibrational wavepackets formed on the ground state of a polyatomic molecule – this is important because it provides access to nuclear motion on multiple potential surface, as opposed to the 1-D examinations of prior studies on diatomic molecules; and 3) the quantification of the relative signal amplitudes for competitive excitation channels (i.e. R-type vs. bond softening). It is important to appreciate that the bond-softening mechanism has been proposed, but not experimentally verified, for some time. It is likely that this mechanism has been a contributor to signals measured and reported by many experimentalists, but conventional spectroscopies do not have the needed resolution to unambiguously identify this source; it requires the ultrafast EUV probing capabilities unique to the authors. These experimental data provide the “smoking gun” evidence needed to assign the bond softening mechanism – the $-\pi/2$ phase shift observed in the oscillatory time-dependent transient signals. In addition to its importance to fundamental molecular dynamics research, these finding also have the potential to impact areas such as analytical chemistry by enabling a new type of polyatomic vibrational spectroscopy.

Specific suggestions for revision.

1. As I detail above, this is an important paper, and there is little room for argument over the data interpretation – the authors got it right. However, I don't think the title accurately reflects the content of the manuscript, nor will it attract the broadest readership. In my opinion, the novelty of this work is the detailed assessment of polyatomic vibrational wavepackets and their excitation mechanism. Therefore, I would encourage modifying the title to explicitly state polyatomic molecules are being studied. I know this is more or less implicit in the use of multimode, but I don't know that the average chemist will be attracted by this title. I encourage more specificity here.

We thank the reviewer for the suggestion to explicitly state in the title that polyatomic molecules are the subject of study in this work. As such, we have slightly modified the title to read, “Elucidating the origins of multimode vibrational coherences of polyatomic molecules induced by intense laser fields”.

2. Figures S7 and S8 are critically important to understanding the mechanisms at play. I encourage modifying Figure 1 of the main manuscript to include the more descriptive elements of Figs. S7 and S8. Again, this will help a broader cross section of readers to understand the manuscript.

We thank the reviewer for this excellent suggestion, which we believe greatly improves the readability of the manuscript. To Fig. 1 we have added panels (b) – (d) of Supplementary Figs. 7 and 8, which show the time-dependent position of the vibrational wave packet, its XUV transition energy, and the spectral first-moment signal.

3. This reviewer is very curious about the possibility to access exotic regions of the potential surface, or modes with different symmetries by using light fields with different polarization states. Have the authors considered how the results might be influenced using circularly polarized light vs. linearly polarized light? Can this be commented on in the manuscript?

In the photon picture, bond softening corresponds to far-off-resonance Raman scattering (see ref. 21 of the manuscript). As such, vibrational modes that are amenable to bond softening should also be Raman-active. In Raman spectroscopy, circularly polarized light can indeed be used to probe systems that exhibit non-symmetric Raman scattering, i.e., systems that possess an additional anti-symmetric anisotropy component of the scattering tensor in addition to the conventional symmetric component. However, Placzek has shown that the vibrational Raman tensor is required to be symmetric when the exciting light is far off-resonance. As such, we do not envision any dependence on whether the incident light is linearly or circularly polarized.

Regarding access to exotic regions of (field-free) potential energy surfaces, we point out that a related strong-field effect – the dynamic Stark shift – has been employed to control photodissociation dynamics that involve nonadiabatic crossings [Science **314**, 278 (2006)]. This work is now cited in the introduction section of the revised manuscript as ref. 18. In addition, ref. 32 of the original manuscript describes the use of light-induced conical intersections to control photodissociation dynamics; this reference is now moved to the introduction section of the manuscript as ref. 19.

4. I do make one recommendation that I think can strengthen the conclusions reached by the authors. An important aspect of this work is their ability to resolve the relative contributions of two mechanisms for exciting vibrational wavepackets (*R*-selective depletion vs. bond softening). I really want to emphasize that this capability really sets the current manuscript apart from any others I've seen on the topic. But, this capability presents a great opportunity. Because the mechanisms should seem to have an excitation field dependence, the authors could consider analyzing how the relative amplitudes respond over a range of pulse energies. It would be very helpful to identify if there are thresholds for these different mechanisms and what molecular parameters set these parameters. The authors could expand the discussion on pages 6-8 to include this analysis.

We thank the reviewer for this excellent recommendation. It is indeed true that the amplitudes of vibrational motion launched by bond softening and *R*-selective depletion should each exhibit a distinct dependence on the laser intensity. The simulated intensity dependence is shown in Ullrich's pioneering work for the case of D₂ (ref. 21), where the bond softening-induced vibrational amplitude is found to scale as I^2 , consistent with the correspondence between bond softening and far-off-resonance Raman scattering, the latter being a two-photon process. At low intensities, *R*-selective depletion plays a minor role because of the limited ionization depletion of the neutral species. The relative contributions are reversed at a threshold intensity of $I_{\text{th}} \sim 2.5 \times 10^{14}$ W/cm², with *R*-selective depletion dominating the response for intensities $I > I_{\text{th}}$.

In the case of CH₃I, due to the negligible displacement between the equilibrium C—I distances between the ground-state neutral and ion species ($\Delta R = 0.6$ pm) and the similarity between the vibrational frequencies (hence, curvatures of the potentials) of the two states ($\Delta\omega/\omega = 0.1$), it is not clear that such an intensity threshold exists. To see this, it would be helpful to consider the extreme limit where both ΔR and $\Delta\omega/\omega$ are zero. In this case, *R*-selective depletion vanishes, and only bond softening-induced wave packets can exist. The above, however, considers only the ion electronic ground state and neglects contributions from strong-field ionization to electronically excited ion states, whose equilibrium distances

and vibrational frequencies could vary significantly from those of the neutral and ion ground states. Future experiments that determine the relative contributions from the two channels as a function of laser intensity should allow one to address if an intensity threshold for CH₃I exists. In fact, such experiments could also elucidate the role of electronically excited ion states in *R*-selective depletion.

In the revised manuscript, we point out the importance of intensity-dependence measurements as future work and include the above considerations in a newly created second paragraph of the Discussion section. The paragraph reads, “The amplitudes of vibrational motion launched by bond softening and *R*-selective depletion should each exhibit a distinct dependence on the laser intensity. In general, bond softening is expected to dominate at lower intensities because of limited ionization depletion of the neutral ground state. In the case of D₂, model calculations reveal that the relative contributions are reversed at a threshold intensity of $I_{\text{th}} \sim 2.5 \times 10^{14} \text{ W cm}^{-2}$, with *R*-selective depletion dominating the response for intensities $I > I_{\text{th}}$. Whether such an intensity threshold exists for CH₃I is unclear, given the negligible displacement between the equilibrium C—I distances between the ground-state neutral and ion species ($\Delta R/R = 0.003$) and the similarity between the vibrational frequencies, and hence, the curvatures of the potentials of the two states ($\Delta\omega/\omega = 0.1$). In the extreme limit where both ΔR and $\Delta\omega/\omega$ are zero, *R*-selective depletion vanishes and only bond softening-induced wave packets can exist. In such cases, it is interesting to note that strong-field ionization to higher-lying electronically excited ion states could still result in vibrational motion launched by *R*-selective depletion. Future experiments that determine the relative contributions from bond softening and *R*-selective depletion as a function of laser intensity should allow one to address if an intensity threshold for CH₃I exists and if electronically excited ion states participate in *R*-selective depletion.”

5. Related to comment 4, on page 3 the authors state that bond-softening is valid for low-frequency laser fields. What is “low-frequency” in this context? Again, I am very interested in the relationship between frequency and excitation field strength. Are these “low frequencies” likely to be molecule or symmetry specific? Understanding these limits are important for the development of applications in vibrational spectroscopy.

Bond softening can be understood within either the quasi-static or the Floquet framework. The quasi-static picture is presented in the main manuscript and the associated Fig. 1b, whereas the Floquet picture is briefly discussed in the Supplementary Information. Both perspectives yield the same outcome: the intense laser field distorts the neutral ground-state potential, hence launching vibrational motion.

The statement about low-frequency laser field pertains to the comparison between the timescale on which the laser electric field oscillates with the timescale on which electrons – and hence, the electronic potential – respond to the oscillating laser field. The former is determined by the carrier frequency of the laser pulse, whereas the latter is dictated by the presence of electronic resonances. The low-frequency limit is reached when the laser carrier frequency is much lower than any electronic resonance frequency, a condition fulfilled by the use of a far-off-resonance laser pulse. In this case, the electronic response of the molecule is faster than the timescale on which the laser electric field oscillates, and one can think of bond softening in terms of the adiabatic deformation of the potential energy surface by the laser field, i.e., the quasi-static picture of bond softening.

The low-frequency limit is valid in our experiments because the 786-nm carrier wavelength of the laser pulse is strongly detuned away from the lowest-energy electronic transition of CH₃I, centered at ~260 nm. Deviation from the low-frequency limit occurs for molecules that exhibit absorption resonances in the vicinity of the laser carrier wavelength; in that sense,

what is “low frequency” is molecule-specific, as the reviewer points out. To the best of our knowledge, however, there is no dependence of the low-frequency limit on the symmetries of either the electronic or vibrational states.

6. Finally, can the authors comment on how general the bond softening mechanism might be. Depletion methods are easy to understand, but is the bond softening mechanism limited to specific ground state potentials? Can we expect this method to be applicable to molecules beyond halogen-containing species?

As mentioned in our reply to point #3 raised by the reviewer, vibrational modes that are Raman-active should also be susceptible to bond softening. This defines the generality of bond softening. To emphasize this, we have inserted the following sentence into p. 10 of the revised manuscript: “Since bond softening corresponds to far-off-resonance Raman scattering within the photon picture, vibrational modes that are Raman active are expected to be susceptible to bond softening.”

Indeed, we believe that bond softening is applicable to molecules beyond halogen-containing species. As explained on p. 10 of the original manuscript, bond softening-induced vibrational motion is expected for molecules for which valence ionization results in the removal of a nonbonding electron. Aside from HCl, H₂O, and organic halides, which are already listed in p. 10 of the original manuscript, other possible (non-halide) candidates include CO₂ and alcohols. We now add these other candidate molecules to the list on p. 10 of the revised manuscript, along with relevant references that report the investigations of strong-field phenomena in these molecules.

Reviewer #2 (Remarks to the Author):

This is a beautiful manuscript, which presents results on using time resolved XUV absorption to track vibrational wave packets launched by a strong field laser pulse. The results are very nice and the analysis and discussion thorough and enlightening.

At first I was somewhat skeptical of the results, because it seemed to me that the sensitivity of the measurement to displacements claimed by the authors was not possible. However, I went through the analysis in detail, and given the signal to noise and sensitivity of the authors to small changes in the spectrally resolved absorption with time delay as well as the large shift in the absorption spectra with displacement, it indeed seems like it is possible to achieve the sensitivity to small displacements that the authors claim.

I was also a little skeptical of the extraction of the phase information for each of the modulation frequencies, but the SI gives a nice detailed discussion of this which I found convincing.

I found the discussion of why the authors find a different amount of bond softening vs R dependent ionization from earlier results very nice and informative. However, I think it would be good to have some discussion of the intensity dependence of their measurements. How do the pump probe measurements vary with pump pulse intensity?

We thank the reviewer for the suggestion to include some discussion on the possible intensity dependence of the results. This suggestion is identical to comment #4 raised by Reviewer #1. Hence, we would like to direct the attention of the reviewer to our reply to the earlier comment. As requested by this reviewer and Reviewer #1, we have added a paragraph on the pump-intensity dependence to the Discussion section of the revised manuscript.

While the authors here focus on the measurement of bound state vibrational wave packets, which have been studied with many other approaches, I feel that the sensitivity they achieve and the potential for using their approach to study a broader class of wave packets merits publication in Nature Communications.

So, in conclusion, I think that this is a very well written and exciting manuscript which demonstrates an exquisite sensitivity to vibrational motion via transient XUV absorption spectroscopy. I believe it deserves to be published in Nature Communications once the authors add some discussion of the pump pulse intensity dependence.

REVIEWERS' COMMENTS:

Reviewer #1 (Remarks to the Author):

The authors have addressed all of my comments clearly, and in thorough detail. I support publication of this manuscript in the current form.

Reviewer #2 (Remarks to the Author):

I think that while it would have been nice for the authors to actually have measured the intensity dependence, having not already done so, this would clearly involve new measurements, which would take significant effort and time. The response to both referees is very satisfactory and I think that the manuscript is strong enough without the additional intensity dependent measurements to be published as is. I therefore recommend publication.

Response to reviewers

Reviewer #1 (Remarks to the Author):

The authors have addressed all of my comments clearly, and in thorough detail. I support publication of this manuscript in the current form.

Reviewer #2 (Remarks to the Author):

I think that while it would have been nice for the authors to actually have measured the intensity dependence, having not already done so, this would clearly involve new measurements, which would take significant effort and time. The response to both referees is very satisfactory and I think that the manuscript is strong enough without the additional intensity dependent measurements to be published as is. I therefore recommend publication.

We are grateful to both reviewers for the helpful comments that they have given and for supporting the publication of our work.